# Clinical Characteristics and Distribution of Pediatric Fractures at a Tertiary Hospital in Northern France: A 20-Year-Distance Comparative Analysis (1999–2019)

**DOI:** 10.3390/medicina58050610

**Published:** 2022-04-28

**Authors:** Faustine Monget, Marco Sapienza, Kathryn Louise McCracken, Eric Nectoux, Damien Fron, Antonio Andreacchio, Vito Pavone, Federico Canavese

**Affiliations:** 1Pediatric Orthopedic Surgery Department, Lille University Center, Faculty of Medicine, Hôpital Jeanne de Flandre and University of Lille, Avenue Eugène Avinée, 59037 Lille, France; faustine.mon@gmail.com (F.M.); marcosapienza09@yhahoo.it (M.S.); 120105096@umail.ucc.ie (K.L.M.); e_nectoux@hotmail.fr (E.N.); damien.fron@chru-lille.fr (D.F.); 2A.O.U. “Policlinico—Vittorio Emanuele” P.O. V. Emanuele—Clinica Ortopedica Pad. 1, Via Plebiscito, 6284 Catania, Italy; vitopavone@hotmail.com; 3School of Medicine, University College Cork, College Road, T12 K8AF Cork, Ireland; 4Pediatric Orthopedic Surgery Department, “Buzzi” Children’s Hospital, Via L. Castelvetro 32, 20154 Milano, Italy; prof.andreacchio@gmail.com

**Keywords:** epidemiology, fractures, traumatology, children, time trend

## Abstract

*Background and objectives*: The epidemiology and distribution of pediatric fractures change over time and are influenced by a multitude of factors including geography, climate, and population characteristics. The aims of our work were to study the distribution of traumatic pediatric orthopedic injuries admitted to the Lille University Hospital (LUH) Pediatric Emergency Department in 1999 and in 2019 and to analyze the epidemiological differences 20 years apart. *Materials and methods:* This was a retrospective, comparative, monocentric, and epidemiological study involving all children between 0 and 15 years and 3 months of age who consulted the pediatric emergencies of LUH from 1 January 1999 to 31 December 1999 and from 1 January 2019 to 31 December 2019. On admission, the following data were collected: sex, age at the time of injury, month and time of the day the trauma occurred (4:00 a.m to 11:59 a.m, 12:00 p.m. to 19:59 p.m, and 20:00 p.m to 3:59 a.m.), mechanism of injury, laterality (right or left), anatomical location, type of injury, and whether the fracture was closed or open. The type of treatment (orthopedic or surgical) was collected from the medical records. *Results:* A total of 939 children were included in 1999 compared with 781 in 2019 (21% decrease); the average age of children with fractures was significantly higher in 1999 (8.81 years) than in 2019 (7.19 years). This difference was explained by the majority involvement of older children (10–15 years) in 1999 (43% of fractures in 1999 versus 25% of fractures in 2019). Conversely, small children (1–5 years) had significantly more fractures in 2019 (36%) than in 1999 (24%). *Conclusions:* Overall, the types and sites of fractures did not change over the studied time despite a change in the population and mechanism of injury. This suggested that the reflexes of breaking a fall still tended to implicate and damage the same bone segments. Finally, the proportion of fractures managed surgically versus orthopedically has not evolved since 1999. Exploring this is a possible area of further research that would complement our study.

## 1. Introduction

The epidemiology and distribution of pediatric fractures change over time and are influenced by a multitude of factors including geography, climate, and population characteristics such as lifestyle and demographics [1,2,3]. During the past two decades or so, dramatic demographic and socio-cultural modifications have influenced the epidemiology of pediatric fractures. At the same time, fracture management options have also evolved due to technical advances and the further development of surgical knowledge [4].

The emergence of new leisure devices such as hoverboards, electric scooters, trampolines, and monkey bars has led to changes in the daily activities of the pediatric population with a possible change in traumatic mechanisms. In addition, the physical and extracurricular activities offered to schoolchildren have further evolved with the feminization or masculinization of several sports [5,6,7,8,9,10].

Similarly, the progress of prevention protocols and systems as well as increased access to dedicated pediatric emergency facilities have further influenced both the epidemiology and the distribution of fractures in pediatrics as well as their treatment [1,2,3,4].

This article used the trauma registry of Lille University Hospital (LUH) to identify the data of all children younger than 15 years and 3 months of age admitted to the Pediatric Emergency Department for orthopedic traumas during the period from 1 January–31 December 1999 and from 1 January–31 December 2019. 

The aim of our work was to study the distribution of traumatic pediatric orthopedic injuries admitted to the LUH Pediatric Emergency Department in 1999 and in 2019 and to analyze the epidemiological differences 20 years apart based on gender, age, mechanism of injury, seasonal variation, treatment (orthopedic or surgical), and anatomical region. The hypothesis was that significant changes occurred during this 20-year period.

## 2. Materials and Methods

Lille is a first-tier city in Northern France with a population of 212,597 and 233,098 inhabitants in the years 1999 and 2019, respectively. The greater urban area of Lille is the sixth largest in France and covers a population of >3,000,000 inhabitants. The percentage of children younger than 15 years of age was 15.7% in both 1999 and 2019 [10,11]. LUH is a 3222-bed tertiary hospital in Lille and it is the Regional Medical Center for Children of Northern France. There were 19,300 and 29,734 pediatric emergency department visits in 1999 and 2019, respectively.

After Institutional Review Board approval, medical records were reviewed to identify all children aged 0 to 15 years and 3 months admitted to the Pediatric Orthopedic Department of LUH for orthopedic traumas during the two time periods evaluated in this study.

The inclusion criteria were as follows: (1) admission through the Pediatric Emergency Department of LUH during the period from 1 January 1999 to 31 December 1999 and 1 January 2019 to 31 December 2019; (2) age less than 15 years and 3 months at the time of injury; (3) reason for admission: orthopedic trauma; and (4) complete clinical and radiographic data.

Patients with incomplete medical records and/or imaging data were excluded from the analysis. In addition, patients in which the mechanism of injury could not be specified were excluded.

On admission, the following data were collected: sex, age at the time of injury, month and time of the day the trauma occurred (4:00 to 11:59 in the morning; 12:00 to 19:59 in the afternoon; and 20:00 to 3:59 at night), mechanism of injury, laterality (right or left), anatomical location, type of injury, and whether the fracture was closed or open. The type of treatment (orthopedic or surgical) was collected from the medical records.

### Statistical Analysis

A statistical analysis was performed using the Statistical Package for the Social Sciences (SPSS) version 25.0 for Windows. The data were evaluated using descriptive statistics to determine the frequency of the different factors surveyed. Comparisons between the groups for different nominal variables were made using chi-squared or Fisher exact tests. A *p*-value < 0.05 was considered to be statistically significant.

## 3. Results

The total number of pediatric patients admitted through the Pediatric Emergency Department of LUH for any reason during the year 2019 was 29,734; this represented a 35% increase over the total number of admissions in 1999 (*n* = 19,300). Specifically, 7797 out of 19,300 (40.4%) and 2825 out of 29,734 patients (9.5%) were admitted for orthopedic traumas in 1999 and in 2019, respectively (*p* < 0.05). Among the patients admitted for orthopedic traumas, 939 were diagnosed with a fracture in 1999 (12%) and 781 in 2019 (27.6%) (*p* < 0.05).

### 3.1. Month of Injury

The breakdown of fractures showed an even distribution of injuries throughout both study periods. In 1999, the month with the highest incidence of fractures was May (*n* = 150, 16%) and the lowest was July (*n* = 42, 4.5%) whereas in 2019, June accounted for the highest frequency of fractures (*n* = 108, 13.8%) and February was the lowest (*n* = 32, 4.1%) (Figure 1) [10]. 

### 3.2. Age

The average age of the occurrence of fractures in children was significantly higher in 1999 (8.81 ± 4.12 years) than in 2019 (7.19 ± 4.00 years) (*p* < 0.001).

A comparison of the results broken down by age group highlighted these results. 

The percentage of fractures among patients aged 11–15 years was significantly higher in 1999 (405/939; 43%) than in 2019 (195/781; 25%) (*p* < 0.001) (Table 1).

In contrast, there were more fractures in the 1–5-year age group in 2019 (281 patients; 30%) than in 1999 (224 patients; 24%).

### 3.3. Mechanism of Injury

The primary mechanism responsible for pediatric fractures in 1999 was a fall, accounting for 27% of fractures. Similarly, it remained the most common mechanism in 2019, but increased to be responsible for 41% of all fractures. 

The second most frequent mechanism of injury in 1999 was a direct trauma, accounting for 25% of cases. It remained at this position in 2019, but with a lower percentage (12.8%) (Table 2). 

It was interesting to note that, regarding sports activities, group sports appeared to be responsible for more fractures in 1999 whereas individual sports accounted for the majority of sports injuries in 2019. This result could indicate a development in individual physical activity over the last 20 years. Finally, MVA were responsible for a greater proportion of fractures in 1999 (6.4%) compared with 2019 (1.4%) (Table 2).

### 3.4. Fractures Treated Surgically

In 1999 and 2019, the majority of surgically treated fractures were due to falls (30%), collective sports (25%), and a direct trauma (17%).

In 1999, the majority of fractures treated surgically occurred in 11–15-year-old patients (45%); only 17% of these fractures occurred among 1–5-year-old patients. In 2019, the age distribution for surgically treated fractures was fairly homogenous: 31% (1–5-year-old), 38% (6–10-year-old), and 31% (11–15-year-old) (Table 1 and Table 3).

Of the 939 fractures recorded in 1999, 801 (85%) were treated orthopedically and only 138 (15%) received a surgical treatment. In 2019, 83% (648/781) of fractures were treated orthopedically and 17% (133/781) were treated surgically. There were no statistically significant differences between the proportion of orthopedic versus surgical treatments between 1999 and 2019.

### 3.5. Site and Type of Fracture

The site and type of fractures did not change over time. An upper limb accounted for three times the number of fractures than any other site in both 1999 (77%) and 2019 (72%) with wrist fractures (mostly distal radius) representing 55% of fractures in 1999 and 41% in 2019.

There were slightly more ankle and foot fractures in 2019 (18%) compared with 1999 (11%). Similarly, the proportion of elbow fractures also increased from 11% in 1999 to 16% in 2019 (*p* < 0.001) (Table 4).

## 4. Discussion

There have been many socio-cultural and political changes over the last 20 years that have impacted on both the available and preferred activities of children. As a result, there has been a shift in the epidemiology of pediatric traumas and injuries. This shift was prominent in our trauma center with respect to demographics, the population, and the mechanisms of the pediatric fractures. However, overall fractures and the choice of treatment (orthopedic versus surgical) have remained consistent. Despite a 30% increase in the number of consultations in the pediatric emergency department, the number of trauma and fracture cases has decreased by 20% without any change in the demographics of the regional population over the same time period. This suggests that there has been a large increase in medical consultations and a decrease in traumatic injuries and fractures. 

Our research is one of the few studies to compare the epidemiology of pediatric fractures over a 20-year period. 

In 20 years, there has been a 35% increase in the number of consultations for pediatric emergencies whereas the number of consultations for traumas has decreased by 38%.

Part of this difference may be due to the mode of data collection; in 1999, the patients included were those who had attended the pediatric emergency department, but also included direct transfers from peripheral centers that did not go through the emergency department. These transfers are rare and currently involve fewer than 5 patients per month. The demographic changes in the region did not seem to explain this decrease in the fracture rate among children over the 20-year study period.

In the municipality of Lille, the increase in population (+212,597 in 1999 compared with +232,098 in 2019) was mild with a stable percentage of children <14 years of age (15.7%). A similar demographic trend was identified in France (+5,855,448 in 1999 and +5,626,662 in 2019) [11].

The months of May and June were much more prominent in this study. The literature data suggest the possibility of a circannual rhythm in pediatric traumas with an epidemiological peak in June [11]. These data are consistent with our center; a peak period for fractures was seen in both 1999 (May–June) and 2019 (June).

We found a significant decrease in the age at injury over the 20-year period studied. In 1999, the average age of patients with a fracture was 8.81 years compared with 7.19 years in 2019. The 11 to 15-year age group was more represented in 1999 compared with 2019; surgical fractures occurred more frequently in older children in 1999 and in younger children in 2019. From this, we could infer that the preferred activities of older children shifted in favor of more sedentary activities such as the use of mobile phones and screens; at the same time, surgical indications have evolved. Young children (1–5 years) seemed to be more active and may have been more involved in physical activities in 2019 compared with 1999 [1,2,3,4,5,6,7,8,9].

The mechanism of injury was difficult to ascertain because it varied significantly over the 20-year period. For example, 105 patients from 1999 were excluded from the comparative study because the mechanism of injury could not be sufficiently characterized, thus leading to a difficult interpretation of our data. The main mechanism responsible for pediatric fractures were falls in both 1999 and 2019. There were more fractures involving activities “on wheels” in 1999 than in 2019; specifically, roller skating was twice as present in 1999 than in 2019. The literature seems to suggest that the practice of roller skating is more dangerous than scootering [12,13]. 

With regard to sports, collective sports appeared to be the most frequent cause of sports-related fractures in 1999 whereas individual sports represented the majority in 2019. This result could indicate an increase in the development of individual physical activities over the last 20 years.

MVA were five times more common in 1999 than in 2019, likely reflecting the effectiveness of a road prevention policy, which has been the subject of multiple prevention campaigns in the last 20 years, leading to a 50% reduction of road deaths in France over this period (124,524 in 1999 compared with 56,019 in 2019) [14].

The fracture sites had little variation over time; an upper limb was three times more represented than all other fracture sites in the two study periods and these data were consistent with the literature [4,15]. Wrist fractures (mostly distal radius) and forearm fractures accounted for 55% of fractures in 1999 and 41% in 2019. There were more elbow and ankle fractures among children in 2019 (16% and 18%) than in 1999 (11% each) and more distal radius fractures in 1999 (28%) than in 2019 (21%). Khosla et al. found that the incidence of forearm fractures has increased by 32% among boys and 56% among girls over the past 30 years for reasons that are still unclear. These findings were not reflected in our study [16].

In terms of surgically treated fractures, there was an increase in wrist and forearm fractures in older children and elbow fractures in younger children, which could explain this trend. The majority of fractures in 2019 were seen in younger children because the incidence in older children aged 11–15-years markedly decreased from 43% in 1999 to 25% in 2019. This was likely due to the sedentarization of this age group due to the introduction and increased popularity of screen-related activities over the last 20 years. In contrast, fractures in the 1–5-year age group increased, potentially due to the greater involvement of this age group in physical activities. Furthermore, there was a greater representation of fractures from individual sports and sports-related fractures increased overall in proportion whereas injuries from activities “on wheels” have become less frequent. Fractures from MVA have almost disappeared over the last 20 years, which could be attributed to the efficacy of political measures aimed at preventing road accidents introduced over the same time period. Trampoline accidents were not specifically mentioned in the records from 1999, but were responsible for 5.4% of fractures in 2019 (Table 5). Although the indication of surgical treatment did not appear to have changed over time, surgical practices and techniques have evolved. In a study of 6389 children under the age of 16 over a period of 10 years, Cheng et al. [15] found an increase in the number of fractures in adolescent boys without other significant changes in the fracture pattern. In contrast, they found significant changes in surgical management; closed reduction rates increased from 9.5 to 38.7% in distal radius fractures, from 4.3 to 40% in supra-condyloid fractures, and from 1.8 to 22% in diaphyseal fractures of the forearm. The changes in treatment regimens were also accompanied by a proportional decrease in the open reduction rate and the length of stay in hospital from <10 to 38% [17]. Within the category of surgically treated fractures, however, there were significant differences between the two study periods. 

This study was limited by the difference in how the data were collected between the two study periods, particularly concerning the mechanism of injury. In 1999, 45% of older children (>11 years) received a surgical treatment compared with 31% in 2019. In younger children, the percentage of surgical treatments increased to 31% in 2019 from 17% in 1999. Another limitation was inherent from the retrospective analysis of our data. However, all data were collected from the same institution in the same location with stable demographics.

## 5. Conclusions

The types and sites of fractures have not changed over time despite a change in the population and mechanism of injury. This suggested that the reflexes of breaking a fall still tend to implicate and damage the same bone segments. Finally, the proportion of fractures managed surgically versus orthopedically has not evolved since 1999, but the surgical techniques used were not examined. Exploring this is a possible area of further research that would complement our study. 

## Figures and Tables

**Figure 1 medicina-58-00610-f001:**
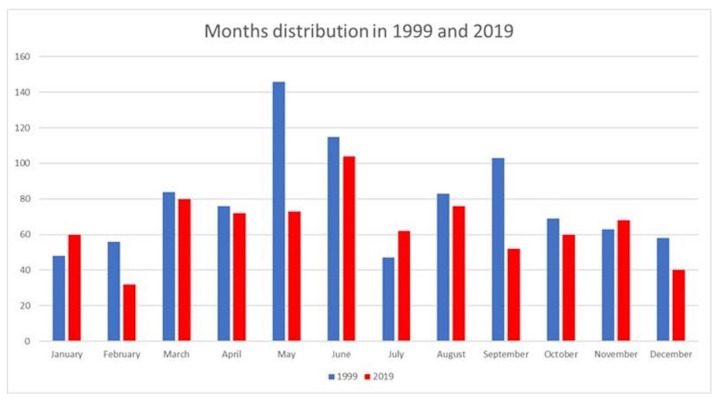
Breakdown of months in which fractures occurred in 1999 and 2019.

**Table 1 medicina-58-00610-t001:** Number of fractures broken down by age group.

		1999 (*n* = 939)	2019 (*n* = 781)	*p*-Value
Age group, n	11–15 years	408 (43%)	195 (25%)	<0.001
	6–10 years	307 (33%)	294 (38%)	<0.001
	1–5 years	224 (24%)	281 (36%)	<0.001
	<1 years	0 (0%)	11 (1.4%)	<0.001

**Table 2 medicina-58-00610-t002:** Mechanism of injury in 1999 and 2019.

		1999 (*n* = 939)	2019 (*n* = 781)	*n*
Mechanism, n	Fall	249 (27%)	319 (41%)	568
	Direct trauma	237 (25%)	100 (13%)	337
	Collective sports	133 (14%)	81 (10%)	214
	Scooter/bicycle	125 (13%)	80 (10%)	205
	Domicile/school/park	114 (12%)	20 (2.6%)	134 (Non interpretable)
	Trampoline	0 (0%)	109 (14%)	110 (Non interpretable)
	Individual sports	20 (2.1%)	61 (7.8%)	81
	MVA	60 (6.4%)	11 (1.4%)	71

(Chi^2^ < 0.001 for all categories.)

**Table 3 medicina-58-00610-t003:** Age of patients at surgical treatment.

		1999 (*n* = 138)	2019 (*n* = 133)	*n*
Age at surgical treatment, n	11–15 years	62 (45%)	41 (31%)	103
	6–10 years	52 (38%)	50 (38%)	102
	1–5 years	24 (17%)	41 (31%)	65
	<1 year	0 (0%)	1 (0.75%)	1

(Fisher < 0.025 for all age groups.)

**Table 4 medicina-58-00610-t004:** Distribution of fractures according to site of injury.

		1999 (*n* = 939)	2019 (*n* = 781)	*n*
Site, n	Forearm/wrist	517 (55%)	319 (41%)	836
	Ankle and foot	102 (11%)	144 (18%)	246
	Elbow	100 (11%)	128 (16%)	228
	Clavicle	74 (7.9%)	78 (10%)	152
	Thigh and knee	49 (5.2%)	38 (4.9%)	87
	Leg	52 (5.5%)	27 (3.5%)	79
	Shoulder and arm	30 (3.2%)	37 (4.7%)	67
	Spine and pelvis	10 (1.1%)	10 (1.3%)	20

(Fischer < 0.001 for all categories.)

**Table 5 medicina-58-00610-t005:** Summary.

		1999 (*n* = 939)	2019 (*n* = 781)	*n*	*p*-Value	Test
Age, mean		8.81 (±4.12)	7.19 (±4.00)	1720	<0.001	Welch
Mechanism of injury, n	Fall	249 (27%)	319 (41%)	568	<0.001	Chi^2^
	Direct trauma	237 (25%)	100 (13%)	337	-	-
	Group sports	133 (14%)	81 (10%)	214	-	-
	Scooter/bicycle	125 (13%)	80 (10%)	205	-	-
	Home/school	114 (12%)	20 (2.6%)	134	-	-
	Trampoline and games	1 (0.11%)	109 (14%)	110	-	-
	Individual sports	20 (2.1%)	61 (7.8%)	81	-	-
	MVA	60 (6.4%)	11 (1.4%)	71	-	-
Site, n	Forearm/wrist	517 (55%)	319 (41%)	836	<0.001	Fisher
	Ankle and foot	102 (11%)	144 (18%)	246	-	-
	Elbow	100 (11%)	128 (16%)	228	-	-
	Clavicle	74 (7.9%)	78 (10%)	152	-	-
	Thigh and knee	49 (5.2%)	38 (4.9%)	87	-	-
	Leg	52 (5.5%)	27 (3.5%)	79	-	-
	Shoulder and arm	30 (3.2%)	37 (4.7%)	67	-	-
	Spine and pelvis	10 (1.1%)	10 (1.3%)	20	-	-
	Elbow	5 (0.53%)	0 (0%)	5	-	-
Treatment, n	Orthopedic	801 (85%)	648 (83%)	1449	0.19	Chi^2^
	Surgical	138 (15%)	133 (17%)	271	-	-
Age group, n	11–15 years	408 (43%)	195 (25%)	603	<0.001	Fisher
	6–10 years	307 (33%)	294 (38%)	601	-	-
	1–5 years	224 (24%)	281 (36%)	505	-	-

## Data Availability

Data available on request due to restrictions (privacy and ethical).

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
