# Peer review of "Clinical Characteristics and Distribution of Pediatric Fractures at a Tertiary Hospital in Northern France: A 20-Year-Distance Comparative Analysis (1999–2019)"

_medicina, 2022, doi:10.3390/medicina58050610_

Round 1
Reviewer 1 Report
Dear Authors
- Abstract - the abbreviation LUH should be explain
- Materials and methods - what does mean- surgical or orthopedic method of treatment?
- Table 3 and 4 - where is p?
- It is necessary to improve Discussion
Author Response
Thank you for contributing to the review of this article.
1.I added the meaning of the acronym LHU in the abstract (Lille University Hospital).
2. Surgical or Orthopaedic means surgical or conservative treatment
3. Tables 3 and 4 are in the manuscript, I’m sorry if you couldn’t see them, could you check again?
4. I am improving the discussion as requested.
Reviewer 2 Report
Dear Authors the topic is very interesting.
By this epidemiological evaluation the Authors underlined the development of pediatric fractures.
As regards the introduction i suggest to cite at line 43 after bibliography n. 2, the following article in which the Authors described the epidemiological data of forearm fracture respect to age, injury mechanism , treatment and complication stratification.
(Failure Predictor Factors of Conservative Treatment in Pediatric Forearm Fractures
Biomed Res Int. 2018; 2018: 5930106.
Maccagnano G et al. )
As regards the M&M, i suggest to include the parameter regarding the complications.
For the reader it will be interesting to know the tendency also of the complications (for instance: % function recovery, rigidity, consolidation delay, refractures……).
Due to this new aspect it will be necessary to improve the results, discussion and conclusion sections
Author Response
Thank you for your important contribution in the review process.
1. I have inserted the required bibliographical reference.
2. I performed the required language editing.
3. Unfortunately, we do not have data on complications as this study refers to the epidemiology of fractures and the data were collected on the basis of access to the emergency room and many patients were followed at other hospitals, therefore it is impossible to derive data on complications in particular those relating to 1999.